# Performance of Tuberculosis Molecular Bacterial Load Assay Compared to Alere TB-LAM in Urine of Pulmonary Tuberculosis Patients with HIV Co-Infections

**DOI:** 10.3390/ijms24043715

**Published:** 2023-02-13

**Authors:** Daniel Adon Mapamba, Elingarami Sauli, Julieth Lalashowi, Joram Buza, Joseph John, Zawadi Mwaisango, Peter Tarmo, Issa Sabi, Andrea Rachow, Nyanda Elias Ntinginya, Bariki Mtafya

**Affiliations:** 1National Institute for Medical Research—Mbeya Medical Research Centre (NIM-MMRC), Mbeya P.O. Box 2410, Tanzania; 2The Nelson Mandela African Institution of Science and Technology, Arusha P.O. Box 447, Tanzania; 3Division of Infectious Diseases and Tropical Medicine, Medical Centre of the University of Munich (LMU), 80802 Munich, Germany; 4German Center for Infection Research (DZIF), Partner Site Munich, 81675 Munich, Germany

**Keywords:** TB-lipoarabinomannan, TB-LAM, urine, tuberculosis-molecular bacterial load assay, TB-MBLA, human immunodeficiency virus, HIV

## Abstract

Alternative tools are needed to improve the detection of *M. tuberculosis* (*M. tb*) in HIV co-infections. We evaluated the utility of Tuberculosis Molecular Bacterial Load Assay (TB-MBLA) compared to lipoarabinomannan (LAM) to detect *M. tb* in urine. Sputum Xpert MTB/RIF-positive patients were consented to provide urine at baseline, weeks 2, 8, 16, and 24 of treatment for TB-MBLA, culture, and LAM. Results were compared with sputum cultures and microscopy. Initial *M. tb.* H37Rv spiking experiments were performed to validate the tests. A total of 63 urine samples from 47 patients were analyzed. The median age (IQR) was 38 (30–41) years; 25 (53.2%) were male, 3 (6.5%) had urine for all visits, 45 (95.7%) were HIV positive, of whom 18 (40%) had CD4 cell counts below 200 cells/µL, and 33 (73.3%) were on ART at enrollment. Overall urine LAM positivity was 14.3% compared to 4.8% with TB-MBLA. Culture and microscopy of their sputum counterparts were positive in 20.6% and 12.7% of patients, respectively. Of the three patients with urine and sputum at baseline, one (33.33%) had urine TB-MBLA and LAM positive compared to 100% with sputum MGIT culture positive. Spearman’s rank correction coefficient (r) between TB-MBLA and MGIT was −0.85 and 0.89 with a solid culture, *p* > 0.05. TB-MBLA has the promising potential to improve *M. tb* detection in urine of HIV-co-infected patients and complement current TB diagnostics.

## 1. Introduction

Tuberculosis (TB) is a global public health problem responsible for high mortality worldwide until the recent COVID-19 pandemic [1]. The World Health Organization (WHO) reported a total of 10.6 million TB cases and 1.6 million deaths in 2021, of whom 0.2 million occurred in people living with human immunodeficiency virus (PLHIV) [1]. Better TB diagnostic and monitoring tools are urgently needed for the detection of TB and clinical management of patients on TB therapy.

Currently, diagnosis of TB relies on sputum smear microscopy, Xpert MTB/RIF-based tests, and culture. Sputum smear microscopy is one of the cheapest tests but has low sensitivity and performs poorly in HIV co-infected patients, extra-pulmonary cases, and pediatrics [2,3]. DNA-based Xpert MTB/RIF assays are very sensitive, but their positivity does not necessarily mean viable TB bacilli [4,5]. Recent reports demonstrate the promising performance of the Xpert MTB/RIF Ultra in pediatric stools and cerebrospinal fluid [6], which will improve the detection of TB in this group of patients and those who cannot produce sputum.

Cultivation of mycobacteria is the most acceptable gold standard for diagnosis and monitoring of TB treatment response but is compromised by loss of data due to contamination, viability loss during sample processing, long time to achieve results, and requires appropriate training and biosafety level 3 laboratory infrastructures which are very expensive to establish in resource-poor countries [2,7]. Furthermore, the positivity yield of culture-based tests when performed in non-sputum samples such as urine and stools is very low [8,9]. Alternative tests that can detect TB in non-sputum specimens, such as urine, will be beneficial to patients with extra-pulmonary TB, including those with advanced HIV infections and children who cannot expectorate sputum samples.

Tuberculosis molecular bacterial load assay (TB-MBLA) is a culture-free test that quantifies the *M. tuberculosis* 16S rRNA as the marker of viable bacilli using a reverse transcriptase quantitative polymerase chain reaction (RT-PCR). A TB-MBLA assay has been demonstrated to be effective, rapid, and accurate when performed in patients’ sputum specimens [10,11]. Moreover, a recent study conducted in Uganda has shown the promising performance of TB-MBLA in stool specimens, and the results were comparable to the Xpert MTB/RIF Ultra test in detecting low bacillary load samples [6]. However, no prior work has investigated the applicability of TB-MBLA for detection of *M. tb* in urine specimens.

In this pilot study, we compared the performance of TB-MBLA to the WHO approved lipoarabinomannan (LAM), a qualitative immunoassay antigen test for the detection of *M. tb* in the urine of HIV co-infected patients, using MGIT liquid culture as the reference standard [12,13].

## 2. Results

### 2.1. Patients’ Characteristics

A total of 47 patients were included in the analysis. The median age (IQR) was 38 (30–41) years; 25 (53.2%) were males; 45 (95.7%) were HIV positive, of whom 18 (40%) had a CD4 cell count <200, and 34 (73.3%) were on ART treatment at the time of enrollment in the study (Table 1, and Figure 1). All patients were bacteriologically positive by Xpert MTB/RIF Assay performed on sputum samples at diagnosis and received standard anti-TB therapy for 2 months consisting of Rifampicin, Isoniazid, Pyrazinamide, and Ethambutol (HZRE), followed by a 4-month continuation phase with Rifampicin and Isoniazid (HR).

#### 2.1.1. *M. tb*, H37Rv Spiking, and Optimization Experiments

A strong Spearman’s rank correction coefficient (r) was obtained between TB-MBLA and solid culture (Middlebrook 7H11), r = 0.89 (Figure 2A), and between TB-MBLA with time to positivity (TTP) in MGIT liquid culture at r = −0.85 (Figure 2B). The correlation between MGIT-TTP and Middlebrook 7H11 media was −0.85 (Figure 2C). The limit of detection (LoD) of MGIT liquid culture was 1.50 × 10^1^ CFU/mL, whilst the LoD of both TB-MBLA and solid culture in Middlebrook 7H11 media was 1.00 × 10^2^ CFU/mL.

#### 2.1.2. Overall Performance of Urine LAM and TB-MBLA Compared to Standard Tests

A total of 63 urine samples were collected from 47 patients; 3 samples were collected at week 0 (baseline), 17 at week 2, 21 at week 8, 19 at week 16, and 3 at week 24 of treatment. Of the 63 urine samples tested, 9 (14.3%) were LAM positive, 3 (4.8%) were TB-MBLA positive, and all urine samples were negative in liquid (MGIT) and solid cultures (Lowenstein Jensen, LJ). Urine LAM positivity was 33.3% at week 0, 23.5% at week 2, 14.3% at week 8, and 5.3% at week 16. TB-MBLA positivity was 33.3% at baseline (week 0) and 11.8% at week 2. Using respiratory sputum specimens as the reference standard for comparison, 13 (20.6%) and 8 (12.7%) of their sputum counterparts were MGIT cultures and ZN smear positive, respectively (Table 2).

#### 2.1.3. Comparison of Urine LAM and TB-MBLA with Sputum-Based Test Results

Three patients (6.5%) out of the 47 patients had both urine and respiratory sputum specimens from baseline to week 24 of anti-TB therapy. At baseline, the percentage (%) positivity of urine TB-LAM and TB-MBLA was 33.3 for each patient, but both TB-MBLA and Alere LAM tests were negative in subsequent follow-up visits in weeks 2, 8, 16, and 24 of anti-TB therapy (Figure 3). Compared to sputum-based tests, three patients (100%) were MGIT and LJ positive, while two patients (66.6%) were smear positive at baseline. The positivity of sputum-based tests declined to 33.3% for MGIT and LJ culture by treatment visit weeks 8 and 24, respectively (Figure 3). There was no difference in the sputum test positivity between smear, culture in LJ, or MGIT at all treatment time points with *p* > 0.05.

## 3. Discussion

Non-sputum-based tests for the detection of *M. tb* such as urine or stool-based assays would improve TB case detection in HIV-co-infected and paediatric patients who frequently fail to produce sputum for standard diagnosis [6,14]. Early studies have demonstrated the higher sensitivity and specificity of the urine Alere LAM test for the detection of TB in patients with advanced HIV infections and children [15,16]. In this pilot study, we show the potential of the TB-MBLA test for the detection of *M. tb* in urine through our in vitro *M. tb* H37Rv spiking experiments and replicated our findings in real patient urine from TB-HIV co-infected patients.

Using *M. tb* H37Rv urine spiking experiments, we were able to detect live *M. tb* and showed a higher correction between TB-MBLA and culture methods, both CFU in solid media and TTP in MGIT liquid culture, reproducing our previously reported results from clinical sputum specimens [10,17]. Furthermore, there was no difference in the amount of detectable *M. tb* bacilli in urine when tested using the TB-MBLA versus solid and liquid media with *p* > 0.05. The limit of detection demonstrated by TB-MBLA, MGIT liquid culture, and solid culture in 7H11 media from urine spiking experiments corresponded to the detection limit previously demonstrated using respiratory specimens [10]. Before applying the test to real patient urine, initial validation experiment was critical for understanding the TB-MBLA’s ability to detect live *M. tb* in urine specimens.

The proportion of patients with urine TB-MBLA positive was higher than the proportion of patients with urine culture positive. Interestingly, the Alere LAM test had the highest positivity compared to both urine TB-MBLA and culture. The higher positivity of the Alere LAM may be explained by the fact that it detects *M. tb* LAM protein secreted in urine, while TB-MBLA and culture detect live *M. tb* organism [18,19,20]. Early studies have shown that LAM proteins with a size of 16 kilodaltons (kDa) can pass through the kidney and be detected in the urine [21]. Moreover, the extent of infections or kidney impairments that would allow the excretion of *M. tb* cells in urine is not yet known, and this may differ from one patient to another and with individual patients’ immune systems. Low CD4 cell counts below 200 cells/uL have been associated with LAM positivity [22,23,24]. However, it is of note that only 37.5% of patients with positive LAM results had CD4 cell counts <200 cells/µL. We postulate that the differences in the concentration of *M. tb* cells and LAM protein measured by TB-MBLA and LAM, respectively, may explain the discrepancy in the performance of the two tests in urine.

Respiratory sputum is the most acceptable diagnostic specimen for pulmonary tuberculosis [25,26]. Using sputum as a diagnostic reference sample, we found a higher positivity rate of sputum culture and ZN smear microscopy compared to urine TB-MBLA or LAM tests. The higher positivity of sputum culture confirms the importance of respiratory samples for the diagnosis of TB in coughing patients and those with the ability to produce sputum specimens. It is plausible to assume that urine may have a lower detectable *M. tb* bacillary load than sputum specimens for coughing patients, which could explain the differences in *M. tb* positivity observed between the two sample types.

It is important to note that both urine and sputum specimens were processed by the NALC-NaOH method before culturing. Initial studies have demonstrated that chemicals used to process samples in order to eliminate non-TB organisms, referred to as “culture contaminants”, have a negative effect on the viability of *M. tb* bacilli and compromise their recovery in culture [17,20]. We believe that the effect of chemicals on *M. tb* viability is consistent regardless of sample type. Thus, low or no *M. tb* yield in urine culture compared to sputum culture may be explained by the low bacillary burden in urine specimens that may have been compromised by chemicals, and the results shifted from positive to negative.

Our study had several limitations. The study was nested in the TB sequel cohort [27], where baseline visits occurred before the commencement of our assessment. Thus, majority of the patients did not have urine samples at baseline, which impacted our sample size, and we were not able to establish the performance of the tests prior to the start of anti-TB therapy. Secondly, most patients had PTB and were on ART treatment before enrollment in the main TB sequel study. Being on ART before enrollment suggests that patients may have already improved their immune systems and consequently impacted the presence or excretion of TB bacteria in urine and compromised test positivity in urine.

In conclusion, our pilot study offers promising results on the potential use of novel RNA-based molecular tests, such as the TB-MBLA, for the detection of viable *M. tb* in urine. Such tests may be beneficial to patients with advanced HIV co-infection, extrapulmonary form of the disease, and pediatrics and complement the current standard tools for diagnosis and monitoring of treatment response. Our future study, with a well-designed sample collection prior to treatment initiation and longitudinal follow-up throughout treatment visits, will provide more insights into the application of TB-MBLA for the detection of *M. tb* in urine.

## 4. Materials and Methods

### 4.1. Study Design and Settings

This was a prospective longitudinal study conducted from February 2019 to August 2020 assessing the performance of urine TB-MBLA compared to Determine Alere LAM test in HIV co-infected with pulmonary TB patients (PTB). Patients with anti-tuberculosis drug resistance as determined by the Xpert MTB/RIF assay or phenotypic drug susceptibility testing in the Bactec MGIT Culture Systems were excluded from the study. The study was carried out in Mbeya Southwest, Tanzania, which is the border crossing point for Malawi and Zambia and has a high TB and HIV burden. Prior to testing real patient urine samples, optimization experiments were performed by spiking *M. tuberculosis* (*M. tb*) H37Rv culture (ATCC 27294) into urine collected from a healthy volunteer. Urine specimens used for initial validation and spiking experiment were *M. tb* negative by Xpert MTB/RIF Assay (Figure 1).

### 4.2. Urine-Spiking Experiment Using M. tb H37Rv Culture

Seven urine aliquots of 3.6 mL were collected into the 15 mL falcon tubes. A total of 0.4 mL of the 0.5 MacFarland standard (approximately 1 × 10^8^ CFU/mL of *M. tb* cells) prepared from a 3-week-old Lowenstein Jensen culture (LJ) was spiked in a urine sample. The samples were serially diluted (10-fold dilutions) in urine up to 1.0 × 10^1^ CFU/mL (*M. tb*). We applied 0.5 mL of each dilution into MGIT liquid media and 0.1 mL into Middlebrook 7H11 media to assess the recovery of viable *M. tb* spiked in urine. The remaining aliquots of 3.0 mL were used for the quantification of viable bacilli by a TB-MBLA test, and three independent replicates were performed for each test (Figure 1).

### 4.3. Collection and Processing of Patient Urine

Urine samples were collected at baseline and throughout treatment weeks 2, 4, 8, 16, and 24 of anti-TB therapy (Figure 1). Urine collection was carried out at the NIMR-MMRC TB clinic and transported to the TB laboratory for processing. Prior to urine collection, patients were instructed to collect mid-stream urine and the required volume for the study. Urine samples were collected into a 60-mL sterile plastic container. A total of 1 mL of urine was mixed with 4 mL of GTC containing 1% Mercapto ethanol in a 15-mL falcon tube immediately after collection to protect RNA from degrading enzymes, as previously described for sputum specimens [10]. The GTC-treated urine, including untreated urine, was transported to the TB laboratory using a cold chain (2–8 °C) for TB-MBLA, culture, and Alere LAM strip test. GTC-treated samples were frozen at −80 °C for batched RNA extraction and quantitative polymerase chain reaction (qPCR).

### 4.4. LAM Determine Urine Test

The lipoarabinomannan (LAM) antigen test was performed on raw urine samples. A total of 60 uL of urine was applied to the LAM Determine strip (Alere) and incubated for 25 min at room temperature. The positivity score and grading were determined using the reference card, following the manufacturer’s guidelines [28], and results were recorded in the laboratory report forms for data entry.

### 4.5. Urine Processing for TB-MBLA

GTC frozen urine samples were thawed prior to RNA extraction. RNA extraction was performed as previously described [20,29,30] using the FastRNA Pro kit (Ambion, BP 50067 Illkirch, France) and removal of the genomic DNA was performed for 1 h using the Turbo DNase Free kit (Ambion, Illkirch, France). The RT-qPCR was carried out on a Qiagen Rotor-Gene PCR machine using the Vital Bacteria Kit (University of St Andrews, College gate, St. Andrews UK) and the TB-MBLA protocol [29]. The RT-PCR cycle quantification (Cq) values were converted to an estimated bacterial load per milliliter (eCFU/mL) of urine using a standard curve pre-developed in the PCR machine from known *M. tb* RNA concentration as previously described [30].

### 4.6. Urine Processing for MGIT Liquid Culture

About 2–5 mL of raw urine was decontaminated by the N-Acetyl-L-Cysteine/Sodium Hydroxide (NALC/NaOH, Sigma) solution for 20 min. Urine pellets were re-suspended in 2 mL of phosphate buffer solution (PBS, pH 6.8, Sigma GmbH, Riedstr., Germany), and 0.5 mL of the suspension was inoculated into MGIT liquid culture Becton Dickinson and company (BD), 7 Loveton Circle Sparks, MD 21152 USA). The tubes were incubated in the BACTEC MGIT 960 System following the manufacturer’s instructions [31]. MGIT cultures were supplemented with oleic acids, albumin, dextrose, catalase (OADC, Sigma GmbH, Germany), and a cocktail of antibiotics consisting of polymyxin B, amphotericin B, nalidixic acid, trimethoprim, and azlocillin (PANTA) following the BD BACTEC MGIT 960 protocol [31]. The purity of culture was confirmed using blood agar plates, and the presence of acid-fast bacilli (AFB) was confirmed by using the Ziehl Neelsen (ZN) stain and *M. tb* speciation by the MPT64 antigen test (BD, 7 Loveton Circle Sparks, MD 21152 USA).

### 4.7. H11 Middlebrook Solid Media

Solid culture in Middlebrook 7H11 was used to assess the viability of *M. tb* spiked in urine samples. A total of 0.1 mL was inoculated into each compartment of the Middlebrook 7H11. The media was supplemented with OADC, 0.5% (*v*/*v*) of glycerol, and a cocktail of antibiotics consisting of Polymyxin, Amphotericin, Carbenicillin, and Trimethoprim (PACT). The plates were incubated at 37 °C for a maximum of 6 weeks. Growth of *M. tb* colonies was observed weekly, and colony count was conducted from a dilution with 10–200 colonies.

### 4.8. Statistical Analysis

The data was entered into a Microsoft Excel spreadsheet and analyzed using GraphPad Prism Software (Version 9.3.1). The correlation between TB-MBLA and time to positivity in MGIT (TTP) or colony count in Middlebrook 7H11 solid media was performed using Spearman’s rank correlation coefficient (r). *T*-test was used to compute the difference between detectable TB bacillary load by TB-MBLA, solid culture in 7H11 and MGIT positivity. The level of significance was assumed at a *p*-value of less than 0.05.

## Figures and Tables

**Figure 1 ijms-24-03715-f001:**
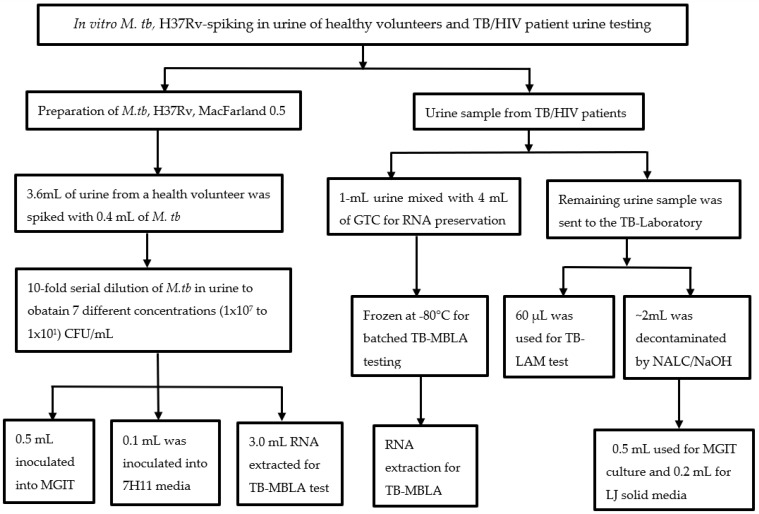
Flow of *M. tb.*, H37Rv spiking experiment, and real patient samples: A 0.5 MacFarland of *M. tb*, H37Rv culture was serially diluted in urine up to 10 × 10^1^ CFU/mL and processed for MGIT, solid culture in 7H11, and TB-MBLA. For real patient urine, 1 mL of fresh urine was treated with guanidine thiocyanate (GTC) for TB-MBLA, 0.06 mL used for TB-LAM, and 2–5 mL decontaminated with the NALC-NaOH method for culture in MGIT and LJ. Three technical replicates were performed for each of the *M. tb* H37Rv spiking experiments.

**Figure 2 ijms-24-03715-f002:**
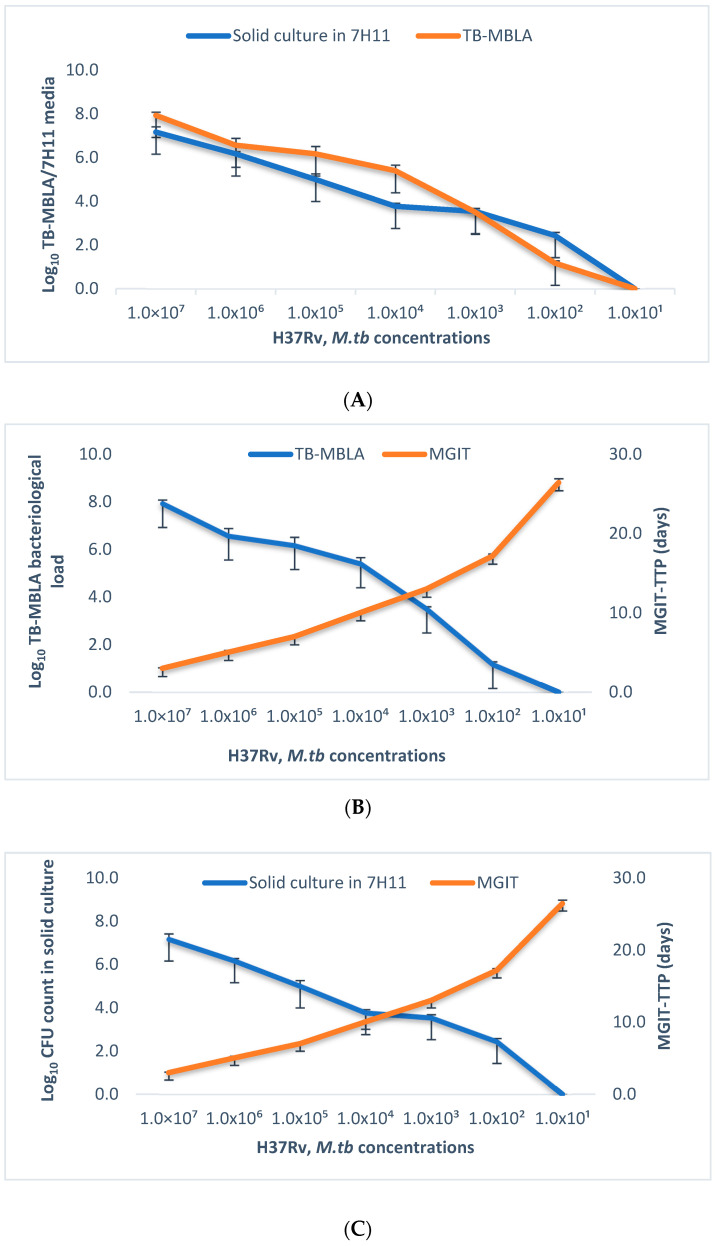
In vitro *M. tb* H3Rv urine spiking experiments. TB-MBLA correlated strongly with 7H11 media in (**A**) and with time to positivity (TTP) in MGIT in (**B**). The correlation between MGIT-TTP and 7H11 media was pronounced in (**C**). Each dot represents average data from three technical replicates of experiments presented with a standard deviation (SD).

**Figure 3 ijms-24-03715-f003:**
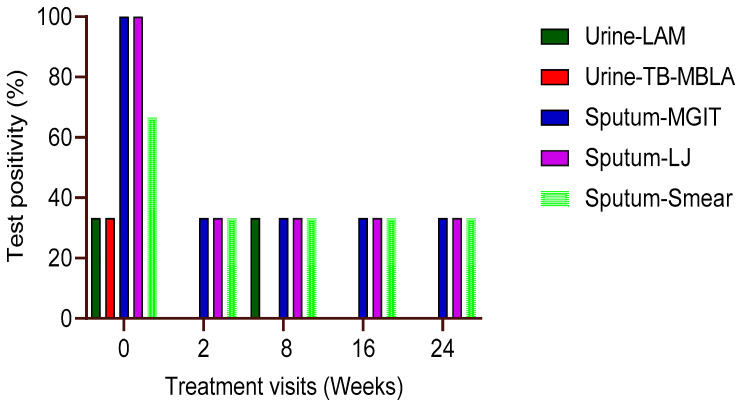
*M. tb* positivity in three patients with all visit samples. The probability of urine TB-MBLA andLAM positivity was low compared to their sputum counterparts at baseline and throughout week 24 of treatment (n = 3). The relation of sputum positivity between each test (smear, MGIT, or LJ) was not different at each time point in treatment, *p* > 0.05.

**Table 1 ijms-24-03715-t001:** Patients’ characteristics at baseline.

Variable	Baseline n (%)
Median age (IQR), years	38 (30–41)
Sex	
Male	25 (53.2)
Female	22 (46.8)
HIV status	
Positive	45 (95.7)
Male	24 (53.3)
Negative	2 (04.3)
Male	1 (50.0)
CD4 count	
<200	18 (40)
≥200	27 (60)
Started ART at D0	12 (26.7)
On ART before D0	33 (73.3)
Xpert MTB/RIF	
High	19 (40.4)
Male	9 (47.4)
Medium	10 (21.3)
Male	6 (60)
Low	16 (34.0)
Male	7 (43.8)
Very low	2 (4.3)
Male	1 (50.0)
ZN Smear grade	
Negative	8 (17.0)
Male	5 (62.5)
Scanty	25 (53.2)
Male	14 (56.0)
1+	4 (8.5)
Male	3 (75.0)
2+	7 (14.9)
Male	1 (14.3)
3+	3 (6.4)
Male	2 (66.7)
MGIT culture	
Positive	43 (91.5)
Male	23 (53.5)
Negative	2 (4.3)
Male	1 (50.0)
Contaminated	2 (4.3)
Male	1 (50.0)

Majority of patients were male, HIV positive, and bacteriologically positive at enrollment by sputum Xpert MTB/RIF Assay, smear, or culture tests (n = 47).

**Table 2 ijms-24-03715-t002:** Performance of TB-MBLA and LAM.

	Urine					Sputum		
Treatment Visit (Weeks)	Total Samples	LAM Positive (%)	TB-MBLA Positive (%)	7H11 Culture Positive	MGIT Culture Positive	MGIT Culture Positive (%)	LJ Culture Positive (%)	Smear Positive (%)
0	3	1 (33.3)	1 (33.3)	0	0	3 (100)	1 (33.3)	2 (66.7)
2	17	4 (23.5)	2 (11.8)	0	0	NTD	NTD	NTD
8	21	3 (14.3)	0 (0)	0	0	6 (28.6)	4 (19.0)	6 (28.6)
16	19	1 (5.3)	0 (0)	0	0	2 (10.5)	1 (5.3)	0 (0.0)
24	3	0 (0)	0 (0)	0	0	1 (3.3)	1 (33.3)	1 (33.3)
Total	63	9 (14.3)	3 (4.8)	0	0	13 (20.6)	7 (11.1)	9 (14.3)

Urine TB-MBLA was positive only at weeks 0 and 2, while LAM positivity was observed up to week 16 of treatment. All urine cultures were negative for *M. tb* cultures, whereas sputum-based smear and culture were positive in a small proportion of patients throughout treatment. Abbreviations: NTD; test not performed because sputum was not collected); LAM, lipoarabinomannan protein; MGIT, Mycobacterium Growth Indicator Tubes; LJ, Lowenstein Jensen, TB-MBLA, Tuberculosis Molecular Bacterial Load Assay; 7H11, Middlebrook 7H11 solid media.

## Data Availability

The data presented in this study are available in the article.

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
