# Peer review of "Performance of Tuberculosis Molecular Bacterial Load Assay Compared to Alere TB-LAM in Urine of Pulmonary Tuberculosis Patients with HIV Co-Infections"

_ijms, 2023, doi:10.3390/ijms24043715_

Round 1
Reviewer 1 Report
Dear Editor,
Thank you for the opportunity to review the manuscript, and I have major concerns about the sample size. Out of 47 samples/patients, only three samples/patients have been studied by the authors, which is not sufficient to get statistically significant data and support the study hypothesis and conclusion.
Thank you,
Namdev
Author Response
Dear Reviewer,
We thank you for reviewing our work. Please find our response uploaded in Word for your reference.
We look forward to hearing from you.
Have a nice time.

Reviewer 2 Report
Dear authors,
Thank you for submitting your study to the International Journal of Molecular Sciences. Please find below my comments and suggestions.
Abstract – missing the spelling of TB-MBLA, LAM
Introduction – paragraph #3 suggestion: add the (long) time required for culture results.
The correct order/presentation of a scientific manuscript is:
Introduction Methods Results Discussion
Please correct the order – the section Material and Methods should be placed right after the Introduction.
Results section: try to improve the description/reporting. It’s a bit confusing.
Table 1. Decide each sex to provide, the same for HIV status.
Reviewer’s final comment: the author developed a pilot study to compare the performance of two test for TB diagnosis, with extrapulmonary samples (urine). This is a very important and relevant study. Overall, the manuscript looks good. However, some adjustments are needed before considering for publication. I look forward to review this article again. Thanks.
Author Response

(The authors gave the same response as above.)

Reviewer 3 Report
Minor comments
In this research article, the authors compared the performance of TB-MBLA to the WHO-approved Lipoarabinomannan (LAM), a qualitative immunoassay antigen test for detecting M. tb in the urine of HIV co-infected patients using MGIT liquid culture as the reference standard. The manuscript is well written, and the story is easy to follow; the experimental design and data analysis are robust.
Point 1: Few abbrevation were not mentioned in the text such as TB-MBLA, LAM etc.
Point 2: Figure 1 flow of M.tb is clear. However, it would be better to draw a more detailed experimental workflow.
In Table 2 majority of patients were male, HIV positive, and bacteriologically positive at enrollment by Sputum Xpert MTB/RIF Assay, smear, or culture tests would be better to show the % of male and female in terms of these assays.
Point 3: In table 2, Performance of TB-MBLA and LAM, it would be better to remove the % 7H11 culture and MGIT culture results in urine, its zero percent.
Point 4: In figure 3 M. tb positivity. Is there any statistical significance at different time points among sputum MGIT, Sputum Smear, and Sputum-LJ. Overall, I could not fault the experiments or the interpretation.
Good Luck
Author Response

(The authors gave the same response as above.)

Round 2
Reviewer 1 Report
Dear Authors,
Thank you for your response and clearing up my doubts, as well as for incorporating the changes suggested by other reviewers.
Thanks,
Namdev